# Keeping the Ductus Arteriosus Patent: Current Strategy and Perspectives

**DOI:** 10.3390/diagnostics15030241

**Published:** 2025-01-21

**Authors:** Anastasios Chatziantoniou, Filippos-Paschalis Rorris, George Samanidis, Meletios Kanakis

**Affiliations:** 1Department of Cardiology, Onassis Hospital, 17674 Athens, Greece; aschatziantoniou@gmail.com; 2Department of Pediatric and Congenital Heart Surgery, Onassis Hospital, 17674 Athens, Greece; prorris@gmail.com (F.-P.R.); meletis_kanakis@yahoo.gr (M.K.); 3Department of Cardiac Surgery, Onassis Hospital, 17674 Athens, Greece

**Keywords:** ductus arteriosus, congenital heart disease, ductal-dependent circulation, single ventricle physiology, Blalock–Taussig shunt

## Abstract

Patent ductus arteriosus (PDA) continues to be a significant finding among infants, as well as adults. What is widely considered to be a problem though, in a group of patients with congenital heart disease, it is the only lifeline. We will initially study the anatomical and biochemical mechanisms affecting the PDA. The main focus of this review is on ductal-dependent congenital heart disease and biochemical and pharmacological approaches to maintaining ductus arteriosus patency, as well as surgical and interventional options for maintaining circulation. The present review aims to highlight gaps in the knowledge regarding the multifunctional role of ductus arteriosus endothelium and possibly propose a new approach to pharmacological maintenance of ductus arteriosus patency.

## 1. Introduction

In fetal circulation, the ductus arteriosus (DA) is a vascular structure that connects the proximal descending aorta to the pulmonary artery, adjacent to the origin of the left pulmonary artery. It is essential in fetal circulation, as it bypasses pulmonary circulation and diverts nonoxygenated blood toward the placenta to support systemic oxygenation. During birth, after the closure of placental circulation, there is a reduction in pulmonary vascular resistance. The lungs become the main source of oxygenation, leading to the DA becoming obsolete and closing spontaneously. In normal-term infants, it closes in >90% by 48 h and in 100% by 96 h of age [1]. In preterm infants, the chance of a PDA is inversely proportional to the time of gestation [2,3]. In normal newborns, if the ductus arteriosus persists and is left open, it is then termed patent ductus arteriosus (PDA). A patent ductus could cause substantial problems for a newborn child, as it causes left-to-right shunting from the aorta to the pulmonary circulation, causing flooding of the pulmonary vascular bed and, ultimately, pulmonary hypertension. If there is not any congenital heart disease requiring a PDA, the ductus is firstly managed pharmacologically with indomethacin, which induces vasoconstriction and closure, and if that fails, an occluding device can be implanted in the catheterization lab.

Ductal-dependent circulation in congenital heart disease consists of conditions that critically affect either pulmonary or systemic circulation, where the diversion of the blood flow is essential for survival. In these cases, it is essential to keep the DA open with pharmaceutical or interventional means or create a systemic-to-pulmonary connection surgically to keep the patient alive.

## 2. Anatomy and Physiology of Ductal-Dependent Congenital Heart Disease

Ductal-dependent congenital heart disease is a term describing complex congenital heart disease with severe systemic or pulmonary circulation abnormalities. In these cases, the neonates’ pulmonary or systemic circulation is wholly dependent on DA patency for its survival. According to the Cochrane Database of Systemic Reviews, there are three broad categories describing ductal-dependent congenital heart disease, depending on the location of the malformation: (a) congenital heart disease with severe obstruction in the right heart or the pulmonary circulation, (b) congenital heart disease with severe obstruction in the left heart or the systemic circulation, and (c) cardiac anomalies requiring adequate mixing of pulmonary and systemic blood flow, such as transposition of the great arteries (Table 1). In all of these cases, an early diagnosis through fetal echocardiography so as to take appropriate measures postnatally in maintaining DA patency is imperative for the neonates’ survival. If these conditions are not diagnosed promptly, during the early days of life when the DA closes spontaneously, there is a severe hemodynamic deterioration of the neonate with a potentially fatal outcome.

In the first category, the most common conditions are pulmonary atresia and critical pulmonary valve stenosis, either as a solitary condition or as part of a condition like tetralogy of Fallot, and tricuspid valve abnormalities such as atresia, which affect the forward flow of blood through the right heart chambers. In such cases, there is reverse blood flow from the right ventricle to the right atrium, leading to right ventricle hypoplasia (except in the presence of a ventricular septal defect, where it is normal), right atrium dilatation, and right-to-left flow through the foramen ovale. The pulmonary artery trunk may be atretic, dilated, or hypoplastic. In these cases, the pulmonary blood flow is dependent on the DA, and its closure may lead to severe hypoxemia and death. Those conditions are characterized by right-to-left shunting and therefore cyanosis.

In the second category, the most common conditions are critical aortic stenosis, coarctation of the aorta, interrupted aortic arch, or hypoplastic left heart syndrome. There are different findings dependent on the position of the obstruction compared to the aortic arch. In conditions where the obstruction is before the aortic arch, such as left ventricular outflow tract obstruction and critical aortic stenosis, there is left-to-right shunting through the foramen ovale to the right heart, which is dilated due to the increased volume. The DA is responsible for irrigating the whole systemic circulation, and there is right-to-left shunting in the aortic arch. If the obstruction is after the aortic arch, the left heart provides blood until the obstruction and the DA provides blood flow to the descending aorta. Postnatal closure of the DA in such cases causes lower body hypoperfusion, circulatory deterioration, and, ultimately, death.

The third category entails conditions like transposition of the great arteries, where there is no apparent obstruction, but the mixing of oxygenated and nonoxygenated blood is essential for the neonate’s survival.

## 3. Embryology, Histology, and Biochemical Regulation of the Ductus Arteriosus

In normal fetal cardiovascular development, the proximal portions of the sixth pair of embryonic aortic arches persist as the proximal branch pulmonary arteries, and the distal portion of the left sixth arch persists as the ductus arteriosus, connecting the left pulmonary artery with the left dorsal aorta. Normally, the distal right sixth aortic arch loses its connection to the dorsal aorta and degenerates by the 8th week of fetal life [4].

In fetal circulation, it is essential for the fetus’ survival for the DA to remain patent. This is achieved with a multitude of dilating factors, since the DA has an intrinsic muscular tone due to its smooth muscle cells. The main mechanism is intracellular calcium concentration reduction, achieved by two main triggers, depending on the timeline of fetal development.

In the early phase of fetal development, the main trigger is nitric oxide (NO) derived from the endothelium, which acts by activating the cyclic guanosine monophosphate and protein kinase G signaling cascade [5]. Similarly to NO, through cGMP mediation, is the mechanism of action of the family of natriuretic peptides, which are produced mainly in the atria, and there are studies that suggest an anti-remodeling effect in pulmonary arteries’ smooth muscle cells [6]. This is further supported by the correlation between increased brain natriuretic peptide (BNP) levels and poor response to indomethacin treatment in preterm infants with PDA [6,7]. Carbon monoxide has been shown to cause DA vasodilation as well, through the inhibition of CYP450 and endothelin-1 [8,9]. Another possible mechanism that warrants further investigation is the vasodilatory mechanism of hydrogen sulfide, which seems to be dose-dependent, as it has been shown by Liu et al., though experimentation in mice DA by Baragatti et al. and in chicken DA by Van der Sterren et al. show a different inter-species response [10,11,12].

Later into pregnancy, as we approach term gestation, the trigger is shifted to prostaglandin E2, mainly originating from the placenta, which interacts with the prostaglandin E2 receptor 4 (EP4) and activates cyclic adenosine monophosphate (cAMP) and the protein kinase A cascade [13,14,15].

After birth, the DA functionally closes at approximately twenty hours after birth and gradually involutes into the ligamentum arteriosum, which is a fibrous structure. Postnatal DA closure can be facilitated by indomethacin, when necessary. Since our aim in this review is maintaining ductal patency after birth, we will focus on the biochemical mechanisms used to normally close DA functionally and anatomically.

As opposed to the mechanisms facilitating patency, the main factor for DA functional closure is an intracellular calcium concentration increase [16,17]. The main trigger in this is the increased O_2_ tension through a plethora of mechanisms. The increase in O_2_ stimulates mitochondrial ATP and H_2_O_2_ production, which, in turn, blocks vasodilating mechanisms like Katp [18]. It also increases the production of 8-iso-prostaglandin F2a, which triggers the Rock (rho-associated protein kinase) cascade, increasing vasoconstriction [6,19], as well as triggering the inositol triphosphate (IP3) signaling cascade, increasing intracellular calcium [20,21]. It is worth noting that H_2_O_2_ further triggers the Rock cascade, leading to a synergistic effect. Lastly, O_2_ further promotes intracellular calcium increase by cytochrome P450 binding of endothelin-1 to endothelin receptor A, leading to IP3 pathway activation [8,12].

There are pathways unrelated to oxygen as well, with the most prominent being glutamate, which promotes DA vasoconstriction through a glutamate receptor (GluR1) that mediates noradrenaline production. This function has been mostly studied in the rat DA by Fujita et al., showing glutamate-induced DA contraction [22]. Corticosteroids, in combination with indomethacin, have also been shown to induce DA closure, probably due to PGE2 sensitivity attenuation [5,23].

The anatomical closure of the DA comes later, and it is mainly achieved through vascular remodeling, a well-studied mechanism, especially in atherosclerosis. It starts with the separation of the endothelial cells from the internal elastic laminae and the proliferation of undifferentiated smooth muscle cells. There is a plethora of factors inducing smooth muscle cell migration and proliferation that we will study below.

Prostaglandin E2 induces smooth muscle cell migration through the Epac pathway, a different mechanism than the protein kinase A cascade, which induces patency, showing a paradoxical effect. PGE2 maintains functional patency but induces remodeling, which, in turn, promotes anatomical closure [24,25].

Another factor, promoting smooth muscle cell cytoskeleton adhesion in the extracellular matrix and inducing vascular remodeling, is transforming growth factor β1 (ΤGF-β1). Tannenbaum et al. investigated its effect in fetal lamb DA smooth muscle cells, increasing a5b1 integrin on the cell surface and promoting cell migration but anchoring the cytoskeleton. This combination mainly leads to smooth muscle cell migration inhibition, though in vitro studies show the opposite effect. This factor is regulated by pulmonary epithelium, shown to be affected by increased pulmonary blood flow in lambs and potentially cause vascular remodeling and pulmonary hypertension, though its effect as a marker or as an inducing factor is inadequately studied [26,27].

Retinoic acid stimulates the growth of DA smooth muscle cells by decreasing apoptosis and increasing cell nuclear antigen expression, as it has been shown by Wu et al. It has been also shown that retinoic acid induces extracellular matrix filling by increasing the production of fibronectin and hyaluronic acid when administered maternally in preterm rats, as studied by Yokoyama et al. [28,29].

Histone modifier gene PRDM6 has been also shown to affect smooth muscle cell proliferation. PRDM6 promotes smooth cell differentiation, as well as induces endothelial cell apoptosis and inhibits endothelial cell proliferation [30,31]. A mutation in PRDM6 in mice has been proven by Li et al. to cause non-syndromic isolated PDA [32].

Lastly, Notch system signaling seems to play an important role in DA remodeling, since the loss of the receptors in smooth muscle cells has been shown to lead to patent ductus arteriosus [33]. This has also been supported by evidence in mice, showing that it was paramount for contractile smooth muscle cell differentiation [34]. Moreover, a pathway involving a Notch signaling inhibitor called γ-secretase inhibitor (DAPT) has been shown to prevent proliferation and migration of DA smooth muscle cells in mice by Wu et al. [35]. Vascular endothelial growth factor (VEGF) is closely related to Notch system signaling and acts as a regulator for neovascularization. It promotes endothelial cell proliferation and DA closure, induced mostly by tissue hypoxia [36,37,38]. This effect is not solely met in DA cells, having an effect on bronchi cells, leading to various lung diseases such as pulmonary arterial hypertension [39,40]. This may reveal a connection between lung and ductus arteriosus epithelium and a potential mechanism for maintaining ductal patency.

## 4. Diagnosis of Patent Ductus Arteriosus and Ductal-Dependent Congenital Heart Disease

As in most congenital heart disease, especially in a group as severe as the one with ductal-dependent circulation, a timely diagnosis is of utmost importance in order to have the optimal outcome [41,42].

Fetal ultrasound is a well-established imaging modality in developed countries. Screening in mid-gestation is recommended as the standard practice, though it has limitations. These consist of a very steep learning curve, as well as technique restrictions such as fetal position and limited ability to diagnose malformations in vascular structures even in very detailed fetal echocardiograms. In congenital heart disease, though a fetal echocardiogram is the imaging modality of choice, there is a significant variation in prenatal detection rates [43]. This variation has been noticed even in developed countries, as suggested by a study in United States showing an interstate range of prenatal diagnosis for congenital heart disease from 11.8 to 53.4% [44]. In underdeveloped countries, it is safe to assume that, where fetal echocardiogram is available, this discrepancy is much more severe.

Recent studies have proposed new echocardiographic approaches for diagnosing ductal-dependent congenital heart disease in fetal life, either by identification of reverse flow in the aortic arch or the ductus arteriosus or through a deep learning artificial intelligence software that processes acquired ultrasound images with results close to those of experienced sonographers [45,46,47].

After birth, the neonate presents with symptoms such as respiratory distress, congestive heart failure, systemic hypotension, shock, or collapse. These depend on the etiology of the ductal-dependent circulation. Early signs consist of cyanosis in ductal-dependent pulmonary circulation and a weakened or absent femoral pulse in ductal-dependent systemic circulation.

These provide useful information for the doctor, and even in the case of no fetal echocardiogram, or in cases where it was not accurately detected prenatally, these symptoms and signs may lead to a timely diagnosis and improved outcomes.

Cardiac ultrasound remains a useful and widely applicable tool, and it is the cornerstone of antenatal congenital heart disease diagnosis. Limited accuracy and a steep learning curve in congenital heart disease are some of the limitations, but in experienced hands, it can provide useful and vital information. It is also easily accessible and readily available in all clinical scenarios. This information consists of general thoracic and abdominal situs, cardiac malformations, valvular or myocardial dysfunction, and cardiac and pulmonary artery and vein anatomy. In ductal-dependent congenital heart disease, in particular, useful information about the underlying pathology causing ductal dependency and its severity, as well as the hemodynamical significance, are provided. The ductus arteriosus physical characteristics can also be assessed, though this is mainly used in patient groups with patent ductus arteriosus without other cardiac conditions. This assessment includes ductus arteriosus morphology and its origin and insertion in the aortic and pulmonary vessels, as well as its size. At this point, it should be noted that size has a poor correlation with the shunt magnitude, though shunt velocity measurement may provide some useful information.

Invasive angiography is the gold standard in these conditions, with known limitations. Cardiac ultrasound remains a useful and widely applicable tool in diagnosis, though it provides limited information in size, width, and hemodynamic significance of a patent ductus arteriosus. Cardiac MRI provides much important information in congenital heart disease in general but is not widely used in this group of patients. This is due to hemodynamic instability, as well as the exam requiring lots of time and usually sedation. There have been recent studies proposing new cardiac CT modalities that require less time and provide anatomical information for small structures like the ductus arteriosus [48].

In patients with ductal-dependent circulation awaiting surgery under continuous prostaglandin infusion, it is of utmost importance to have an experienced pediatric cardiologist in order to be able to diagnose a possibly deteriorating ductus arteriosus. The imaging modality of choice is an ultrasound and requires extended experience in monitoring the patent ductus arteriosus. Daily imaging is also required, as well as continuous patient monitoring and the assessment of any change in the clinical status, in order to prevent DA closure.

## 5. Pharmacological Maintenance of DA Patency and Its Limitations

In ductal-dependent congenital heart diseases, it is imperative for the newborn’s survival to maintain DA patency. The usual aim of this pharmacological intervention is to maintain patency and not inhibit its growth until the infant can undergo surgical repair of its cardiac condition [49,50,51]. The standard of care at this point is PGE1, which promotes vasodilation [52]. It binds to the EP4 receptor, increasing intracellular cAMP, which inhibits myosin light chain kinase, causing vasodilation. The initial dosage of PEG1 is 0.03–0.05 μg/kg/min, which is titrated depending on the infant’s oxygenation [53]. When there is no improvement, the dosage should be increased by 0.05 μg/kg/min increments, as stated by Chamberlin and Lozynski [54], with an aim of ductal patency within thirty minutes to two hours after continuous PGE1 infusion. Many studies suggest PGE1 treatment weaning when optimal oxygenation is achieved in order to reduce side effects, which have been proven to be dose- and treatment length-dependent [55,56,57,58]. The most common side effects of PGE1 consist of apnea, irritability, fever, leukocytosis, convulsions, and hypokalemia, while it may also lead to bradycardia and cardiac arrest in rare cases. Regardless of the dosage, PGE1 may not always have successful outcomes, while, as described by Cucerea et al., the side effects are dosage- and length of treatment-dependent, though mostly transient and treatable [59].

## 6. Patent Ductus Arteriosus Stenting

Since its conception in the early 1990s by Gibbs et al., big steps have been made in arterial duct stenting [60]. Though the early results were considered discouraging, increased expertise and available technology have made ductal stenting a solid, reliable option in palliative therapy in ductal-dependent pulmonary circulation in congenital heart disease. According to the American Heart Association guidelines, it is reasonable to stent an anatomically suitable DA in an infant with a not totally DA-dependent pulmonary blood flow (class IIa, level of evidence B), while it might be reasonable to stent an anatomically suitable DA in an infant with a wholly DA-dependent pulmonary blood flow (class IIb, level of evidence C).

Recent studies have shown the increased use of DA stenting vs. aortopulmonary shunt, especially in cases with high perioperative risk, avoiding neonatal surgery and cardiopulmonary bypass and providing symmetrical pulmonary artery growth. Ductal stenting has also been shown to be superior to an aortopulmonary shunt in terms of reduced need for hospitalization and intensive care stay and non-inferior regarding the survival rate for next-stage surgery [61,62,63].

In cases with ductal-dependent systemic circulation, and especially in hypoplastic left heart syndrome, a different approach where bilateral pulmonary artery bands and an intravascular ductal stent were placed was proposed. This procedure was first described in part by Gibbs et al. in 1993 [64] with poor results initially and was gradually evolved to its contemporary form with better results by Galantowicz et al. [65,66].

All the aforementioned are procedures with good results but which have restrictions in regard to patient weight and growth. This makes the need to maintain ductal patency in the first few days of life even more important.

## 7. The Blalock–Taussig Shunt

The history of systemic-to-pulmonary artery shunts began almost 80 years ago with the inception of an anastomosis between a branch of the aorta (a systemic artery) and the pulmonary artery, first described by Alfred Blalock and Helen Taussig, for improving systemic oxygen saturation in cyanotic congenital heart disease [67]. Since then, various other systemic-to-pulmonary anastomoses have been described, but the initial Blalock–Taussig (BT) shunt has persevered throughout the decades.

In the original BT shunt, an anastomosis between the right subclavian artery and the right pulmonary artery was created after extensive mobilization and distal ligation of the right subclavian artery, usually through a lateral thoracotomy approach. Currently, the modified Blalock–Taussig (MBT) shunt is the preferred choice for a systemic-to-pulmonary artery shunt [68]. In the MBT shunt, a synthetic tube graft made by a 3.5 mm expanded polytetrafluoroethylene (ePTFE, thin-walled Gore-Tex) is used to create a connection between the aorta (central shunt) or the innominate artery (peripheral shunt). The shunt can be created either through a right lateral thoracotomy or with a traditional sternotomy approach. After opening the pericardiac sac, the innominate artery and the right pulmonary artery are completely mobilized. Firstly, an end-to-side anastomosis between the ePTFE graft and the innominate artery is created. After the anastomosis is completed, heparin is administered (100 IU/kg of body weight), and the other end of the graft is anastomosed in the right pulmonary artery in an end-to-side fashion. The size of the shunt is always 3.5 mm for infants of 2.5–3.5 kg and 4 mm for larger infants. The size of the graft is small in order to avoid flooding of blood in the lungs and, subsequently, pulmonary edema. After completion of the anastomosis, the patent ductus arteriosus is divided if needed.

A thrill felt over the pulmonary artery, a rise in arterial oxygen saturation, and a drop in diastolic blood pressure are indicative of adequate shunt function. The systemic-to-pulmonary artery shunts were designed to mimic the role of the ductus arteriosus.

Morbidity and mortality after a MBT shunt remain high, with 13.1% and 7.2%, respectively, as reported in a large retrospective study using The Society of Thoracic Surgeons Congenital Heart Surgery Database [69]. Factors affecting postoperative morbidity and mortality were low weight (<2.5 kg), prematurity, preoperative mechanical ventilatory support, and a diagnosis of pulmonary atresia with intact ventricular septum. Interestingly, the use of cardiopulmonary bypass or concurrent closure of the ductus arteriosus did not affect postoperative mortality.

## 8. Novel Pharmacological Approaches in Maintaining Ductal Patency in Animal Specimens

After studying the current interventional methods of maintaining ductal patency or creating a shunt, it is safe to assume that more study is needed in the field of ductal-dependent circulation, especially in the pharmacological approach (Table 2).

Apart from prostaglandin treatment, with its efficiency and limitations well established, there have been several agents currently found to maintain ductal patency in animal studies.

Milrinone, a phosphodiesterase 3 inhibitor currently in use mostly in heart failure, increases cAMP levels and induces DA relaxation without increasing intimal thickening in newborn rats, as studied by Ichikawa et al. A nonselective endothelin receptor antagonist, TAK-044, has also been found to prevent DA constriction in studies in rats [70].

Another study in rat specimens showed the promising effects of BNP in preventing DA closure. BNP has already been shown to affect pulmonary vasculature, inducing anti-remodeling effects, and it has been shown by Jwu-Lai Yeh et al. to also affect postnatal closure of the DA, maintaining luminal patency, attenuating intimal thickening, and preserving the DA diameter. The proposed mechanism as shown in vitro was through the cGMP pathway, where its anti-remodeling effects were shown to affect the DA as well [71].

Cinaciguat, a soluble guanylyl cyclase activator, works similarly through the cGMP/PKG pathway, having anti-remodeling and vasodilatory effects. Cinaciguat has mainly been studied in pulmonary hypertension patients having these same effects. In a study by Yu-Chi Hung et al., cinaciguat has been studied in neonatal rat specimens. In vivo, it prevented DA closure, maintaining luminal patency and reducing intimal thickening. Ex vivo, cinaciguat prevented oxygen-induced DA constriction dose-dependently. Moreover, it inhibited angiotensin II-induced reactive oxygen species production, as well as DA smooth muscle cell proliferation and migration [72].

One of the most promising molecules is γ-secretase inhibitor DAPT, which inhibits Notch pathway signaling and induces anti-remodeling effects on the DA, possibly through decreasing the calcium influx, reduced reactive oxygen species production, and gene expression downregulation. It is important to note, though, that these were in vitro findings in rat DA epithelium, highlighting the need for further studies [35].

Finally, a recent study revealed a new factor called exendin-4, which is a glucagon-like peptide-1 receptor agonist (GLP-1RA) that can help maintain ductal patency in postnatal rats through vasodilatory and anti-remodeling effects. GLP-1RA has been shown to attenuate pulmonary hypertension through the cAMP/PKA pathway and has similar pharmacological and physiological properties as GLP-1 but has a longer half-life. Exenidin-4 has been proven to maintain ductal patency by attenuating oxygen-induced vasoconstriction and reducing intimal thickening. This factor proposes a novel mechanism of maintaining ductal patency in neonates, though further investigation is imperative [73].

## 9. Future Perspectives

As stated above, there are numerous pathways that affect DA closure or patency, as well as many pharmacological agents that could induce a breakthrough in the current approach.

Considering the type of patients (infants, low body weight, and prematurely born), a consistent study of novel methods is extremely difficult. As previously discussed, almost all of these studies have been carried out with animal specimens, and taking into account the well-studied use of prostaglandin in maintaining ductal patency, further studies need an extremely careful approach.

In the process of reviewing the bibliography, our research group has noticed a particular connection that has not been studied consistently before. This connection is between the pulmonary vasculature epithelium and the ductus arteriosus epithelium, where many of the aforementioned pharmacological agents seem to have the same effect. This is prevalent, especially in the cGMP and GLP-1R/PKA pathway and the BNP, cinaciguat, and exenidin-4, which have been proven to affect pulmonary vasculature in pulmonary hypertension, and recent studies show a similar effect in DA cells [71,72,73]. This connection could provide a novel approach not only in maintaining DA patency but also in the treatment of pulmonary hypertension by pooling the collective knowledge and resources of both conditions.

## 10. Conclusions

Keeping the DA patent remains the only therapeutic short-term option for a selected group of patients. Through this review, we aimed to shed some light on a probable future approach to maintaining ductal patency in ductal-dependent congenital heart disease. There are many steps needed to allow consistent research, especially in this group of fragile patients, but there is also a significant benefit to be had. This benefit consists of keeping the patients in the best possible condition and them reaching an adequate body weight in order to have optimal results through the inevitable corrective surgery. We also established a connection between the pulmonary and the ductus arteriosus epithelium with possible common biochemical pathways that have not been studied adequately so far. This not only opens up a new horizon for research but also a possible cooperation between different fields of study. What is certain is that there is fertile ground for research in this group of patients, and the more we understand about the biochemical correlations in human physiology, the more such connections will be established.

## Figures and Tables

**Table 1 diagnostics-15-00241-t001:** Types of duct-dependent circulation.

	Duct-Dependent Pulmonary Circulation	Duct-Dependent Systemic Circulation	Duct-Dependent Systemic and Pulmonary Circulations
Most common conditions	Pulmonary atresia Critical pulmonary stenosis Tricuspid atresia Tetralogy of Fallot	Coarctation of the aorta LVOT obstruction Hypoplastic left heart syndrome Critical aortic stenosis Interrupted aortic arch	Transposition of the great arteries
Symptoms upon closure	Hypoxemia, cyanosis, desaturation	Systemic hypoperfusion, acidosis, circulatory deterioration	Both of the aforementioned

**Table 2 diagnostics-15-00241-t002:** Novel therapeutic factors in maintaining DA patency (factors marked with * have a similar effect on pulmonary epithelium).

Therapeutic Factor	Biochemical Pathway	Main Mechanism of Action
Milrinone [70]	Phosphodiesterase 3 inhibitor Increases cAMP levels	DA relaxation without increasing intimal thickening
TAK-044 [70]	Endothelin receptor antagonist	DA constriction prevention
γ-secretase inhibitor DAPT [35]	Notch pathway signaling inhibition	DA anti-remodeling effects
BNP * [71]	cGMP pathway	DA smooth muscle cell anti-proliferation and anti-remodeling effects DA vasodilation
Cinaciguat * [72]	cGMP/PKG pathway	DA anti-remodeling effects DA vasodilation (dose dependent)
Exendin-4 * [73]	GLP-1RA antagonist/ cAMP-PKA pathway	DA anti-remodeling effects DA vasodilation

Abbreviations: DA, ductus arteriosus; cAMP, cyclic adenosine monophosphate; BNP, B-type natriuretic peptide; cGMP, cyclic guanine monophosphate; PKG, protein kinase G; GLP-1RA, glucagon-like peptide-1 receptor agonist.

## Data Availability

Not applicable.

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
