# Peer review of "Keeping the Ductus Arteriosus Patent: Current Strategy and Perspectives"

_diagnostics, 2025, doi:10.3390/diagnostics15030241_

Round 1
Reviewer 1 Report
Comments and Suggestions for Authors
In this review the authors described pathophysiology and menagement of patent ductus arteriosus. They focuse on the multifunctional role of ductus arteriosus endothelium and reported sufficient findings regarding treatment approaches on this issues. Although the findings are impressive, I would like to make some comments.
1. Cardiovascular and respiratory consequences of postnatal DA require to be discuss in the section Introduction along with appropriate statistical data
2. The authors should widely discuss the underlying mechanisms of postnatal DA closure and impcat on this several drugs including PGs.
3. Hystological findings along with endothelium-dependent regulation of DA closure are likely to be thoroughly described.
4. Section Conclusion: please, make it clearer than it is adding concise findings
Author Response
Reviewer 1 Comments
Comment 1: Cardiovascular and respiratory consequences of postnatal DA require to be discuss in the section Introduction along with appropriate statistical data
Answer 1: Thank you for your comment. New information were added in the introductions sections relating to the cardiovascular consequences of postnatal DA.
Comment 2: The authors should widely discuss the underlying mechanisms of postnatal DA closure and impcat on this several drugs including PGs.
Answer 2: Thank you for your comment. The biochemical mechanisms of DA function are widely discussed in Section 3. The focus of this review was maintaining the patency of DA and thus we did not discuss its closure mechanisms. However, following up on your comment, a sentence was now added in Section 3 regarding DA closure.
Answer 3: Hystological findings along with endothelium-dependent regulation of DA closure are likely to be thoroughly described.
Comment 3: Thank you for your comment. Extensive histological findings and underlying endothelium properties are discussed in Section 3.
Answer 4: Section Conclusion: please, make it clearer than it is adding concise findings
Comment 4: As this paper was a narrative review and not a study it was difficult for us to provide concise findings in the conclusion section. We made slight adjustments.
Reviewer 2 Report
Comments and Suggestions for Authors
The article is well conceived, well presented.
Interesting subject.
The article refers to an important problem related to congenital cardiac pathology.
It is well known that in fetal circulation, the ductus arteriosus (DA) is a vascular structure that connects the proximal descending aorta to the pulmonary artery, adjacent to the origin of the left pulmonary artery. The role is essential in fetal circulation, as it bypasses pulmonary circulation and diverts nonoxygenated blood toward the placenta to support systemic oxygenation. During birth, after the closure of placental circulation, the lungs become the main source of oxygenation, leading to the DA becoming obsolete and closing spontaneously. In normal-term infants, it closes in >90% by 48 hours and in 100% by 96 hours of age. In preterm infants, the chance of a patent DA is inversely proportional to the time of gestation.
Ductal-dependent circulation is essential for survival in some congenital heart disease so in these cases, it is essential to keep the DA open with pharmaceutical or interventional means or create a systemic-to-pulmonary connection surgically to keep the patient alive.
The article is well structured, well conceived, the authors initially describe the anatomical and biochemical mechanisms affecting the PDA. The main focus of this review is on ductal-dependent congenital heart disease, biochemical and pharmacological approaches to maintaining ductus arteriosus patency, as well as surgical and interventional options for maintaining circulation.
Also the article aims to highlight a new approach to pharmacological maintenance of ductus arteriosus patency after the presentation of the classical pharmacological methods of maintenance of DA patency and its limitations.
Related to the Patent ductus arteriosus stenting even though early results were considered discouraging, increased expertise and available technology have made ductal stenting a solid, reliable option in palliative therapy in ductal-dependent pulmonary circulation in congenital heart disease. Recent studies have shown the increased use of DA stenting vs aortopulmonary shunt, especially in cases with high perioperative risk, avoiding neonatal surgery and cardiopulmonary bypass and providing symmetrical pulmonary artery growth. Ductal stenting has also been shown to be superior to aortopulmonary shunt in terms of reduced need for hospitalization and intensive care stay, and non-inferior regarding survival rate for next-stage surgery. In cases with ductal-dependent systemic circulation, and especially in hypoplastic left heart syndrome, a different approach where bilateral pulmonary artery bands and an intravascular ductal stent were placed was proposed. All these procedures had good results but with restrictions in regard to patient weight and growth. This makes the need to maintain ductal patency in the first few days of life even more important.
The Blalock-Taussig shunt is associated with a high morbidity and mortality so, novel pharmacological approaches in maintaining ductal patency are needed.
After the presentation of the current interventional methods of maintaining ductal patency or creating a shunt, it is safe to assume that more study is needed in the field of ductal-dependent circulation, especially in the pharmacological field.
Apart from prostaglandin treatment, with its efficiency and limitations well established, there have been several agents currently found to maintain ductal patency in animal studies, such as: Milrinone, a phosphodiesterase 3 inhibitor currently in use mostly in heart failure, increases cAMP levels and induces DA relaxation, TAK-044 a nonselective endothelin receptor antagonist, BNP, γ-secretase inhibitor DAPT, Cinaciguat, a soluble guanylyl cyclase activator, works similarly through the cGMP/PKG pathway, having antiremodeling and vasodilatory effects, Exenidin-4 - a glucagon-like peptide-1 receptor agonist (GLP-1RA). Future Perspectives are also discussed.
Pertinent conclusions.
Many references are old and very old and some of them with a far relation with the article.
I have marked them with colour and I suggest to review the references.

Author Response
Comment 1: Many references are old and very old and some of them with a far relation with the article.
Answer 1: Thank you for your comment. We are reviewed the marked references. Indeed, some of them are old but we would like to leave them unchanged as many of these are landmark papers in our understanding of ductus arteriosus.